# Insights into the Resistome and Phylogenomics of a ST195 Multidrug-Resistant *Acinetobacter baumannii* Clinical Isolate from the Czech Republic

**DOI:** 10.3390/life11101079

**Published:** 2021-10-13

**Authors:** Patrik Mlynarcik, Monika Dolejska, Iva Vagnerova, Jana Petrzelova, Iva Sukkar, Veronika Zdarska, Milan Kolar

**Affiliations:** 1Department of Microbiology, Faculty of Medicine and Dentistry, Palacky University Olomouc, 77515 Olomouc, Czech Republic; Iva.Vagnerova@fnol.cz (I.V.); nika.zdarska@gmail.com (V.Z.); milan.kolar@fnol.cz (M.K.); 2Department of Biology and Wildlife Diseases, Faculty of Veterinary Hygiene and Ecology, University of Veterinary Sciences Brno, 61242 Brno, Czech Republic; monika.dolejska@gmail.com; 3CEITEC VETUNI Brno, University of Veterinary Sciences Brno, 61242 Brno, Czech Republic; kutilova.iva@gmail.com; 4Department of Clinical Microbiology and Immunology, Institute of Laboratory Medicine, The University Hospital Brno, 62500 Brno, Czech Republic; 5University Hospital Olomouc, 77900 Olomouc, Czech Republic; jana.petrzelova@fnol.cz; 6Institute of Molecular and Translational Medicine, Faculty of Medicine and Dentistry, Palacky University Olomouc, 77900 Olomouc, Czech Republic

**Keywords:** *β*-lactamase, antibiotic resistance, PCR, *Acinetobacter baumannii*, bacteria

## Abstract

Increasing antimicrobial resistance in nosocomial pathogens, such as *Acinetobacter baumannii*, is becoming a serious threat to public health. It is necessary to detect *β*-lactamase-producing microorganisms in clinical settings to be able to control the spread of carbapenem resistance. This study was conducted to evaluate the presence of *β*-lactamases in a selected clinical isolate of *A. baumannii* of ST2^P^/ST195^Ox^ and to characterize possible enzymes, as well as its *β*-lactam resistome, using PCR and whole-genome sequencing analysis. PCR and sequencing confirmed that the isolate harbored five *bla* gene alleles, namely, *bla*_ADC-73_, *bla*_TEM-1_, *bla*_OXA-23_, *bla*_OXA-58_ and *bla*_OXA-66_, as well as aminoglycosides, macrolides, sulfonamides and tetracyclines resistance determinants, which were either chromosomally and/or plasmid located. Furthermore, a gene order comparison using MAUVE alignment showed multiple changes compared with the clinical isolate of Malaysian *A. baumannii* AC30 genome and 76 regions with high homology. This study suggests that resistance to *β*-lactams in this *A. baumannii* isolate is mainly due to an overproduction of *β*-lactamases in combination with other resistance mechanism (efflux pump system).

## 1. Introduction

The emergence of carbapenem resistance in *Enterobacteriaceae* (e.g., *Escherichia coli*, *Klebsiella pneumoniae*) and Gram-negative non-fermenting (GNNF) bacteria (e.g., *Acinetobacter baumannii*, *Pseudomonas aeruginosa*) is a serious public health concern, mainly due to the production of carbapenemases, which ultimately limit therapeutic possibilities [1].

During evolution, bacteria have developed various genetic and biochemical mechanisms that allow them to resist the strong inhibitory effects of antimicrobials. At present, the most serious and feared mechanism leading to a rapid increase in resistance is considered to be the property of bacteria to exchange plasmids carrying genes encoding *β*-lactamases.

Several *β*-lactam resistance mechanisms have been described in bacteria, including enzymatic inactivation by *β*-lactamases, alterations in antibiotic binding sites, changes in cell permeability, modifications of metabolic pathways and an overexpression of efflux pumps [2]. Regarding the multi-drug resistance (MDR) phenotype in *A. baumannii*, many studies describe that it is related to the production of natural *β*-lactamases, reduced cell permeability and/or hyperexpression of efflux pumps [3]. Furthermore, it was found that the MDR phenotype in *A. baumannii* was significantly attributed to a resistance island carrying transposons, integrons and other mobile genetic elements, and also that efflux systems can provide resistance to a particular antibiotic to a lesser extent than the inactivating antibiotic enzymes [4].

According to the EARS-Net [5], the resistance of *Enterobacteriaceae* and GNNF bacteria to carbapenems was extremely different. For example, in 2019, *Acinetobacter* spp. showed a 31% resistance to carbapenems in the Czech Republic but reached more than 55% in Slovakia. Carbapenem resistance in *Acinetobacter* spp. is currently most critical in the following countries: Croatia (92%), Greece (92%) and Romania (88%) (available at: http://atlas.ecdc.europa.eu/public/index.aspx; accessed on 31 March 2021).

Due to the increasing clinical importance of many *β*-lactamases, their detection is also essential in routine microbiological practice. This study was mainly focused on detecting genes encoding *β*-lactamases, as well as other resistance genes, and associated insertion sequences in a clinical *A. baumannii* strain 11069/A.

## 2. Materials and Methods

### 2.1. Bacterial Strain and Antimicrobial Susceptibility Testing

The *A. baumannii* strain 11069/A (imported from Hungary) was primarily isolated from the eye after a gunshot wound of a patient (man, 61-year-old) staying in the surgical intensive care unit of the University Hospital in Olomouc, Czech Republic.

Antibiotic susceptibility of *A. baumannii* 11069/A was performed using a broth microdilution method and interpreted according to current EUCAST guidelines (V11.0, www.eucast.org; accessed on 31 January 2021). The following antibiotics were tested: cotrimoxazole, meropenem, tigecycline, amikacin, gentamicin, colistin, ciprofloxacin and tobramycin. The isolate was also tested for carbapenemase production using a CARBA NP colorimetric method [6].

### 2.2. Detection of β-Lactamase Genes by PCR 

The presence of *β*-lactamase genes was tested using PCR with genomic DNA that was extracted from the isolate under previously reported conditions [7,8,9,10,11,12,13,14] and using primers shown in Appendix A.

The following strains with known *β*-lactamase genes were used as positive controls for PCR: CTX-M- and TEM-1-positive *E. coli* (NCTC 13400); SHV-positive *K. pneumoniae* (NCTC 13368); IMP-type-producing *E. coli* (NCTC 13476); KPC-3-positive *K. pneumoniae* (NCTC 13438); NDM-1-positive *K. pneumoniae* (NCTC 13443); VIM-10-positive *P. aeruginosa* (NCTC 13437); OXA-23-positive *A. baumannii* (NCTC 13301) and OXA-48-positive *K. pneumoniae* (NCTC 13442).

### 2.3. Whole-Genome Sequencing, Annotation and Molecular Analysis

Genomic DNA from *A. baumannii* isolate 11069/A was extracted and purified using (NucleoSpin® Tissue, Macherey Nagel, Duerren, Germany). Fragment library was constructed using a Nextera XT kit followed by sequencing using Illumina HiSeq according to the manufacturer’s instructions. The raw reads were assembled using a de novo method implemented in Geneious Prime [15] set to medium to high sensitivity of default setting (Biomatters, Auckland, New Zealand).

The Institut Pasteur and Oxford multilocus sequence typing (MLST) scheme (Pasteur (^P^)/Oxford (^Ox^)) (http://pubmlst.org/abaumannii/; accessed on 30 July 2021) [16] was used to type the isolate of interest and was determined from the genome sequence data.

All *β*-lactam resistance genes, as well as other resistant determinants, were identified by automated screening with ResFinder v.4.1 for chromosomal and acquired antimicrobial resistance genes (ARGs) [17], and verified by Geneious Prime using *β*-lactamase genes described in the Beta-Lactamase DataBase (BLDB; http://bldb.eu; last accessed on 31 May 2021) [18].

Plasmid replicon-associated genes were detected using the nucleotide Basic Local Alignment Search Tool (BLASTn; version 2.12.0; http://www.ncbi.nlm.nih.gov/blast/; accessed on 15 April 2021) program.

Open reading frames were captured using the Geneious software analysis tool Glimmer and sequences were searched against the public sequence databases using the BLASTn. Sequence features were annotated using the Geneious Prime annotation transfer tool.

Repeat regions, as well as insertion sequences (IS), were identified and analyzed using Geneious Prime. The predicted open reading frames (ORFs) located upstream and downstream of the ARGs were found using Glimmer and these ORFs were subsequently subjected to BLAST analysis. Predicted ORFs that showed significant sequence similarity to known IS were annotated as IS elements.

### 2.4. Genome Analysis and Phylogenetic Analysis

The genetic map of the assembled genome of *A. baumannii* 11069/A was visualized and compared with *A. baumannii* ST195^Ox^ strain AC30 using Mauve Genome Alignment (MCM algorithm) through the Geneious plugin with the default setting. An *A. baumannii* strain AC30 was used as the reference genome for sequence comparison.

Progressive Mauve with default setting in Geneious Prime was then used to compare the genome of the *A. baumannii* 11069/A strain with the genome sequences of other *A. baumannii* ST195^Ox^ strains downloaded from GenBank. 

A total of 10 *A. baumannii* ST195^Ox^ clinical isolates were used in this work to represent diverse geographical locations, i.e., China (Ab-3, SVUL00000000; 2012046, NDXK00000000), India (AB07, CP006963), Malaysia (AC12, CP007549; AC30, CP007577), Morocco (ABE12_M, FPEF00000000), Lebanon (AC_2355, MJBA00000000), Saudi Arabia (AB263, LYNI00000000), Thailand (T173, JRTY00000000) and the USA (CCF1, LYZL00000000).

Further maximum likelihood phylogenetic tree was generated and visualized using PhyML implemented in Geneious with 100 bootstrap replicates.

### 2.5. Data Availability

Whole-genome assembly was deposited in GenBank under the accession number JAHBBB000000000. 

## 3. Results

### 3.1. Antimicrobial Susceptibility Testing

The isolate showed high resistance to specific antimicrobials and susceptibility was detected only for tigecycline and colistin (Table 1).

### 3.2. PCR Detection of β-Lactamase Genes

One member of the class C *β*-lactamase family (ADC), eight members of class A *β*-lactamases (CARB, CTX-M, GES, KPC, PER, SHV, TEM, VEB), three gene families of subclass B1 (IMP, NDM, VIM) and one member of class D *β*-lactamases (oxacillinases; OXA) were examined using PCR for the selected isolate. Out of the 13 different types of *β*-lactamases examined, 10 types of genes were not experimentally confirmed. The detected *β*-lactamase genes were as follows: *bla*_ADC-like_, *bla*_TEM-like_ and *bla*_OXA-like_ (Appendix A). In case of OXA, however, out of the eleven subfamilies examined, the presence of three subgroups was confirmed. Namely, the genes were *bla*_OXA-23-like_, *bla*_OXA-51-like_ and *bla*_OXA-58-like_. The following primers were used to detect these genes (see Appendix A): ADC-F1/R1, TEM-F/R, OXA(2)-F/R, OXA(5)-F/R and OXA(6)-F/R. The sequence identities with the *β*-lactamase gene sequences of the GenBank database using BLAST analysis were as follows: *bla*_ADC-like_ (100%), *bla*_TEM-like_ (100%), *bla*_OXA-23-like_ (100%), *bla*_OXA-51-like_ (100%) and *bla*_OXA-58-like_ (100%).

### 3.3. Whole-Genome Analysis of MDR A. baumannii 11069/A including Antibiotic Resistance Genes and Associated Genes

A total of 628,395 reads were assembled with the Geneious assembler. *A. baumannii* 11069/A draft genome sequence was assembled into 339 contigs (>201 bp long, base quality scores >Q50).

MLST revealed that the strain 11069/A belonged to the sequence type ST2^P^/ST195^Ox^ (Table 1). Subsequently, genome sequence analysis confirmed that the *A. baumannii* isolate harbored five different *β*-lactamase genes, including *bla*_ADC-73_, *bla*_TEM-1_, *bla*_OXA-23_, *bla*_OXA-58_ and *bla*_OXA-66_. In all cases, a 100% nucleotide identity to the variants of the genes deposited in the BLDB database was confirmed (data not shown).

Our genome analyses demonstrated that the strain 11069/A possessed a total of 13 antibiotic resistance genes, including *tet*(B) (tetracycline resistance), *sul2* (sulfonamide resistance), *msr*(E) and *mph*(E) (macrolide resistance), and four aminoglycoside resistance genes (*aph(3′)-Ia*, *aph(3″)-Ib*, *armA* and *aph(6)-Id*) (Table 2). 

The *bla*_ADC-73_ gene was found to be associated with an IS*Aba1* element (Figure 1-I). Sequencing the data further indicated that *bla*_TEM-1_ was located between two inversely oriented copies of IS*6*-like elements of the IS*26* family with perfect identical terminal inverted repeats of 14 bp on each side (Figure 1-II). This region showed a 100% nucleotide sequence identity to sequences from *A. baumannii* strain WM99c (GenBank accession number, GAN—CP031743). Nonetheless, CP031743 had a deletion of 12,201-bp AbGRI3 resistance region (GAN—KX011026) carrying three ARGs (*armA*, *msr*(E) and *mph*(E)), as well as protein involved in an initiation of plasmid replication (RepB) in the adjacent downstream region compared to the sequence of our isolate. Further, BLASTn was used to detect related plasmid sequences in GenBank. The right end of the contig showed a high level of similarity (56.7% coverage, 100% identity) to a 20,139-bp *armA/msr*(E)*/mph*(E)-carrying plasmid (p2BJAB07104 (GAN—CP003907)) identified in a clinical isolate of *A. baumannii* (Appendix A). In addition, the BLAST results also showed a similar sequence identity (100%) with the (1) *A. baumannii* strain VB2107 chromosome (CP051474), (2) antibiotic resistance island AbGRI3 (KX011026) as well as transposon ∆Tn*6279* chromosomally inserted in strain A072 (KT354507) (Figure 2).

The truncated insertion sequence IS*Aba1* was found upstream of the *bla*_OXA-23_ gene (Figure 1-III). IS*Aba3* family transposase, IS*30* family transposase (both upstream) and an IS*Aba3*-like were identified downstream of the *bla*_OXA-58_ gene (Figure 1-IV). BLASTn analysis showed a high level of similarity (9.5% coverage, 100% identity) to a 29,823-bp *bla*_OXA-58_*/msr*(E)*/mph*(E)-carrying plasmid (pABIR (GAN—EU294228)) (Appendix A). Further, the *aph(3′)-Ia* gene was encoded in a 2,008 bp region flanked by two truncated IS*6*-like element IS*26* family transposase. The two IS*26*-like elements were in direct orientation and no inverted repeats have been identified on both extremities of the IS*26*-*aph(3′)-Ia*-IS*26* fragment (Figure 1-VII). No insertion element was detected upstream of the *bla*_OXA-66_ gene. However, the 2,988-bp region around the *bla*_OXA-66_ gene was identical (100%) to that described in the *A. baumannii* strain CFSAN093706 chromosome (GAN—CP061523). This region included a FxsA cytoplasmic membrane protein, upstream of *bla*_OXA-66._ Downstream of *bla*_OXA-66_, genes encoding N-acetyltransferase and helix-turn-helix transcriptional regulator were found (Figure 1-IX). Several different IS elements (such as IS*Aba1***,** IS*4*-like, IS*5*, IS*91*-like) were also present near the *tet*(B), *sul2*, *msr*(E), *mph*(E), *aph(3′)-Ia*, *aph(3″)-Ib*, *armA* and *aph(6)-Id* (Figure 1). 

Further, an investigation of the mutation in the *carO* gene and efflux pumps involved in carbapenem resistance in *A. baumannii* was performed. Sequencing revealed an altered nucleotide sequence by studying the *carO* outer-membrane protein gene, resulting in a 99.6% (change of one amino acid: glycine to aspartic acid at position 184) amino acid identity, respectively. No deletions or insertions were observed throughout the sequence. The *adeABC* efflux pump, as well as the presence of the *adeRS* genes, were not confirmed. On the contrary, we confirmed two additional resistance-nodulation cell division type efflux systems, namely, AdeIJK and AdeFGH. Further analysis revealed that the tested *A. baumannii* isolate had OprD (OprE3) porin, with no mutation over its coding frame and promoter sequences. In this study, the transcript level of efflux pumps and OprD was not studied. Genomic sequence analysis revealed the presence of the *blc* gene potentially involved in antibiotic resistance, but no *β*-lactamase genes were found in its vicinity.

A genomic comparison between *A. baumannii* strain AC30 (3.8 Mbp) and *A. baumannii* strain 11069/A (3.8 Mbp) genomes was conducted in order to identify genomic islands or regions of genomic plasticity. The Mauve alignment showed 76 regions with high homology named Local Colinear Blocks (LCBs), which encompassed 96.07 and 94.06% of their genomes, respectively (97.1% nucleotide sequence identity) (Figure 3).

### 3.4. Phylogenetic Analysis of A. baumannii 11069/A

The phylogenetic tree diagram obtained using MAUVE and PhyML showed that the 11069/A strain is the most closely related strain to a clinical isolate of Malaysian *A. baumannii* AC30 (Figure 4).

## 4. Discussion

Pan-drug resistance is becoming an increasing threat worldwide [20]. This is mainly related to the fact that more and more carbapenemases are plasmid-encoded and harbored on integrons or transposons. In addition, there are recent concerns about the transmission of *β*-lactamase-producing bacteria in farm animals that have occurred in many European and Asian countries [21,22,23].

In the present study, a clinical isolate of *A. baumannii* with known phenotypic resistance was examined for the presence of most *β*-lactamase genes (mainly the largest and clinically significant groups), which were described only in GNNF bacteria. A PCR, performed using different primer combinations, confirmed the presence of different *bla* genes, specifically *bla*_ADC_, *bla*_TEM_, *bla*_OXA-23-like_, *bla*_OXA-51-like_ and *bla*_OXA-58-like_. Homologous regions in nucleotide sequences of *bla*_PER-like_ and *bla*_SHV-like_ genes were used for designing primers with Primer3 (Geneious Prime). To detect some *β*-lactamase genes, a combination of two pairs of primers was used to cover all or almost all the allelic variants of these genes described in the BLDB database that also served to assess the sequence variant. The PER-F/R primers were aimed at detecting 11 subtypes of the *bla*_PER_ genes but were unable to distinguish four allelic variants (PER-2/-6/-12/-14). The SHV-F/R primers can be used to detect 195 SHV subtypes, while not being able to distinguish five subtypes (SHV-39/-112/-202/-208/-210).

To identify the genes responsible for *β*-lactamase activity in the isolate, short-read WGS analysis was conducted. The analysis revealed that the isolate belonged to ST2^P^/ST195^Ox^, a high-risk clone that is known to be present in various (Eastern) Asian countries and was found to contain carbapenemases [24]. In addition, a carbapenem-resistant *A. baumannii* ST195^Ox^ had been isolated from a Pakistani male living in Denmark and from a Norwegian patient who was hospitalized abroad, suggesting the importation of this clinical strain [25,26]. Further, five *bla* genes were detected in the isolate of *A. baumannii*, including *bla*_ADC-73_, *bla*_TEM-1_, *bla*_OXA-23_, *bla*_OXA-58_ and *bla*_OXA-66_. These genes were found to be co-occurring with other resistance genes, such as *tet*(B), *sul2*, *msr*(E), *mph*(E), *aph(3′)-Ia*, *aph(3″)-Ib*, *armA* and *aph(6)-Id*), together conferring an MDR profile to *β*-lactams, aminoglycosides, macrolides, sulfonamides and tetracyclines. Moreover, in our case, one type of AmpC (ampicillin chromosomal cephalosporinase) *β*-lactamase, *bla*_ADC-73_, was found. Elsewhere, chromosomal-encoded AmpC *β*-lactamase has been reported to have very little activity against carbapenems, but may act synergistically with other resistance mechanisms, such as a diminished production of *oprD*, increased activity of *ampC* and increased expression of several efflux systems [27]. However, no novel *β*-lactamase genes were identified. All the above-mentioned ARGs have been identified in a chromosome and/or plasmid. However, due to some short contigs, such as contig 36 encompassing *bla*_OXA-23_, we were unable to determine precisely whether the gene is chromosomal- or plasmid-encoded.

Truncated IS*Aba1* was detected upstream of the *bla*_ADC-73_ and *bla*_OXA-23_ genes, whereas no insertion sequence was observed upstream of *bla*_OXA-66_. IS*Aba1* confers promoter sequences to the *bla* genes, resulting in *β*-lactamase overproduction in *A. baumannii* [28]. Concerning the *bla*_OXA-58_ gene, this gene was bracketed by insertion sequences (IS*Aba3*-IS*30*-*bla*_OXA-58_-IS*Aba3*-like), both of which have been shown to provide promoter sequences enhancing the expression of *bla*_OXA-58_ [14]. In addition, the *bla*_OXA-58_ gene appears to be plasmid-borne. Poirel et al. reported that the *bla*_OXA-58_ gene is usually plasmid-borne and are associated with insertion sequences [29]. The analysis of the region surrounding the *bla*_TEM-1_ gene revealed two copies of IS*26* in opposite orientations, including inverted repeats, which may play an essential role in transposition (Figure 2). Furthermore, the *aph(3′)-Ia* gene, located between two copies of truncated IS*26* with no duplicated target sequences, showed a 100% identity to sequences from *A. baumannii* strain TP3 (GAN—CP060013). However, this fragment likely resulted from a homologous recombination event. It has been reported that the IS*26* element plays a role in the dissemination of chromosomal- or plasmid-localized resistance genes, including *β*-lactamases, by facilitating their mobilization between different genetic regions [30,31]. Taken together, these genes and the presence of insertion sequences such as IS*Aba1*, IS*Aba3*, IS*Aba3*-like, IS*26* and IS*30* are important in the development of carbapenem-resistant *A. baumannii*.

Study of the *β*-lactam resistome in the isolate of interest confirmed the presence of the *carO* gene. A mutated version of this gene was observed. The type of mutation identified in the porin gene included only substitution (glycine to aspartic acid at position 184; G184D; numbering of amino acid position followed the GAN—CP059354), which probably has little effect on the function of the outer membrane protein. Zhu et al. reported that the CarO mutation is considered to be one of the molecular mechanisms that affect imipenem resistance in *A. baumannii*, indicating that the major differences were amino acid deletion at position 133 and insertion at 140–141 and 154–156 [32]. Furthermore, only one difference was observed at the 48th position (threonine → serine), which had little relation to antibiotic resistance.

Another major outer membrane protein associated with imipenem resistance is OprD [10]. OprD has been reported to be a key porin for the uptake of carbapenems into *A. baumannii* [33]. Our isolate was found to carry the *oprD* gene, whereas no modifications and mutations of this gene were found (the level of gene expression of OprD porin was not investigated). On the other hand, the abnormalities or loss of OprD porin represent one of the major mechanisms of carbapenem resistance in *P. aeruginosa* [34]. Estepa et al. have described many abnormalities in the *oprD* gene, including substitutions, insertions, deletions and premature stop codons. 

Another important antimicrobial resistance mechanism present in *A. baumannii* is hyperexpression of various efflux pumps belonging to the resistance-nodulation cell division family (AdeABC, AdeFGH and AdeIJK) [35]. The *adeABC* operon was not confirmed in this strain. Further, two efflux systems were identified in the *A. baumannii* isolate, namely AdeFGH and AdeIJK, but the level of gene expression of efflux pumps was not investigated here. It has been reported that the efflux pumps AdeABC and AdeFGH are implicated in acquired resistance, while AdeIJK is important for intrinsic resistance [36,37,38]. According to the literature [39], strains containing the AdeIJK pump confer resistance to various antibiotics including carbapenems, whereas AdeFGH has been found to be important, for example, in resistance to chloramphenicol and fluoroquinolones. Therefore, in addition to the presence of IS*Abal*/*bla*_OXA-23_, IS*Abal*/*bla*_ADC-73,_ IS*Aba3*-IS*30*-*bla*_OXA-58_-IS*Aba3*-like, *bla*_OXA-66_ and IS*26*/putative mobile element/*bla*_TEM-1_/IS*26*, these pumps are other important mechanisms that occur in this strain.

In some enterobacteria, the *blc* gene (encoding an outer membrane lipoprotein) has been found to be in the vicinity of the *bla*_AmpC_ gene that encodes a serine *β*-lactamase of chromosomal origin [40,41]. The isolates were characterized by carrying the transposon-like element (IS*Ecp1*-*bla*_CMY_-*blc*-*sugE*) and composite transposon (IS*26*-IS*Ecp1*-*bla*_ACC_-*gdhA*-IS*26*), wherein IS*Ecp1* represents an alternative promoter region regulating the overexpression of the *bla* gene. In our case, we were able to confirm only the presence of the *blc* gene, but there was no *bla*_AmpC_ gene in its vicinity.

To date, the presence of *β*-lactamase genes in *A. baumannii* has been described in numerous papers. However, our findings provide more comprehensive information about the presence of various types of antimicrobial resistance genes, including *β*-lactamases, in a clinical isolate of *A. baumannii*, which may play a role in the development of antibiotic resistance at the clinical level and have important implications for the assessment of risks to human health. In addition, our results showed a match between PCR detection and WGS regarding the types of demonstrated *β*-lactamases, but WGS was able to detect other mechanisms of resistance to *β*-lactams unrelated to *β*-lactamases. Therefore, a PCR assay specifically aimed at detecting some *β*-lactamases in a selected group of microorganisms can be useful for laboratories that do not have next-generation sequencing platforms or do not have sufficient financial resources.

The increasing prevalence of carbapenem-resistant bacteria possessing different carbapenemase genes underscores the implementation of strict control measures to prevent their spread. Further, it is increasingly discussed in the literature that the emergence of resistance is associated with the uptake of resistance determinants throughout the genome. It has been found that the multi-drug resistance phenotype in *A. baumannii* is significantly attributed to the resistance island carrying transposons, integrons and other mobile genetic elements, while it has also been clarified that efflux systems may confer resistance to particular antibiotics to a lesser extent than drug-inactivating enzymes [4].

## 5. Conclusions

In the present study, five different *β*-lactamases and other resistance genes were detected together in an isolate of *A. baumannii*. The insertion sequences, IS*Abal*, IS*Aba3*, IS*Aba3*-like, IS*26* and IS*30*, detected upstream and downstream of the *β*-lactamase genes, may be involved in their dissemination and expression processes. Altogether, these data indicate that the multi-drug resistance to *β*-lactams in this isolate may be a result of the overexpression of the *β*-lactamase genes and other resistance mechanisms (the efflux pump system). The high prevalence of *β*-lactamase-producing *A. baumannii* isolates is of great concern. Therefore, sustained investigation seems essential to monitor *β*-lactamase-producing bacteria in patients with nosocomial and community-acquired infections.

## Figures and Tables

**Figure 1 life-11-01079-f001:**
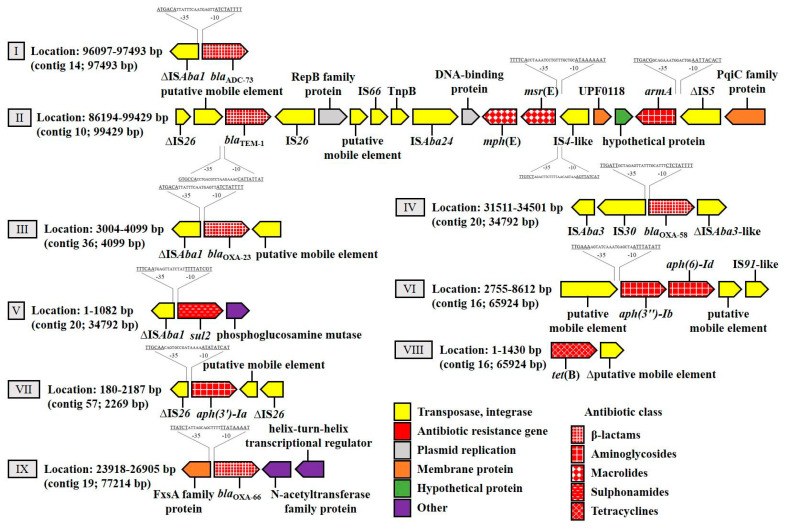
Schematic map of the genomic regions of strain 11069/A encoding antibiotic resistance genes. Arrows showed the direction of transcription. The lengths of the arrows are proportional to the length of the genes or open reading frames. Different patterns for antibiotic resistance genes highlighted in red indicate the antibiotic class to which the respective genetic determinant putatively confers resistance. The genes shown with red color are related to antibiotic resistance. Other abbreviations symbols were Δ, genes that are truncated; IS, insertion sequence; TnpB, IS*66* family insertion sequence element accessory protein TnpB. Promoter prediction was performed using Softberry BPROM webtool [19]. Putative gene annotations were assigned using a BLAST search.

**Figure 2 life-11-01079-f002:**
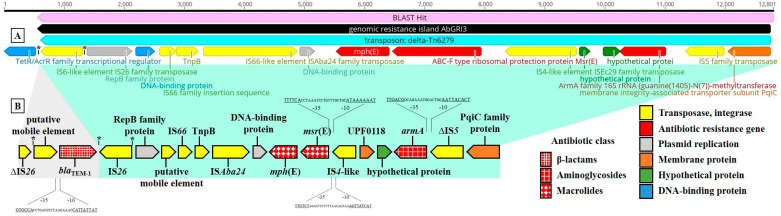
Schematics of genomic islands of the *Acinetobacter baumannii* strains (**A**) and 11069/A (**B**). (**A**) BLAST results showing 100% sequence identity to *A. baumannii* strain VB2107 chromosome (CP051474), antibiotic resistance island AbGRI3 (KX011026) and Tn6279 transposon (KT354507) with one aminoglycoside (*armA*) and two macrolide resistance genes (*msr*(E) and *mph*(E)). (**B**) Genomic island of 11069/A on which the new *bla*_TEM-1_ was found (identical regions are marked in green). The region suggested to be inserted by the IS*26* is marked with gray color. Genes and their transcriptional orientations are shown by horizontal arrows. Inverted repeat sequences are represented by an asterisk (*). Promoter prediction was performed using Softberry BPROM webtool [19]. Putative gene annotations were assigned using a BLAST search.

**Figure 3 life-11-01079-f003:**
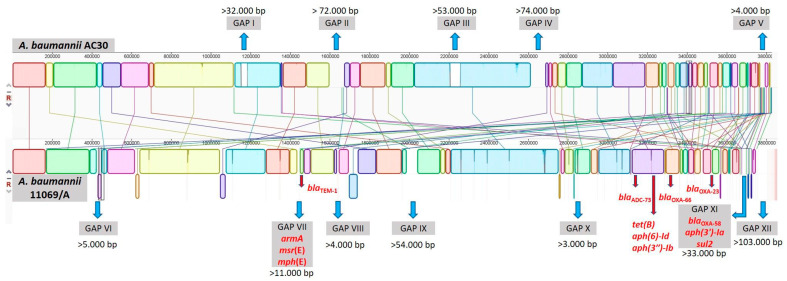
Mauve alignment of *Acinetobacter baumannii* strain AC30 and 11069/A genomes showing areas of missing genes in one relative to the other (genomic islands). LCBs are characterized by blocks of different colors. The white areas indicate similarity gaps in the sequence, and about eight large gaps were identified in *A. baumannii* strains. Completely white regions within LCBs are not aligned and contain sequence elements specific to a particular genome. Antibiotic resistance genes are highlighted in red. More details about distribution of LCBs, presence of gaps and mobile genetic elements are listed in Appendix A.

**Figure 4 life-11-01079-f004:**
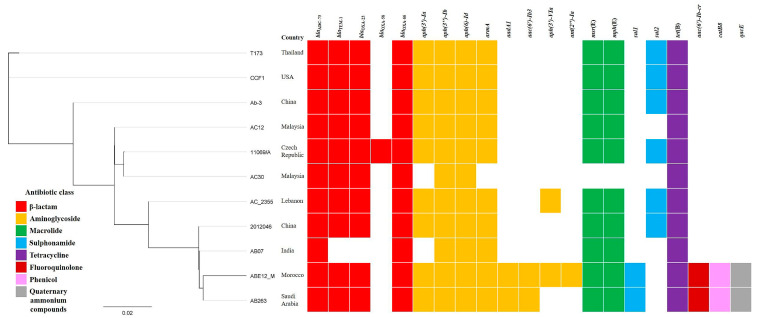
A maximum-likelihood phylogenetic tree showing the *Acinetobacter baumannii* 11069/A in relation to 10 other ST195^Ox^
*Acinetobacter baumannii* strains using PhyML. Phylogenetic analysis was based on 11 genomes of *A. baumannii* ST195 isolates representing diverse geographical locations, i.e., China (Ab-3, GenBank accession number, GAN—SVUL00000000; 2012046, GAN—NDXK00000000), India (AB07, GAN—CP006963), Malaysia (AC12, GAN—CP007549; AC30, GAN—CP007577), Morocco (ABE12_M, GAN—FPEF00000000), Lebanon (AC_2355, GAN—MJBA00000000), Saudi Arabia (AB263, GAN—LYNI00000000), Thailand (T173, GAN—JRTY00000000), the USA (CCF1, GAN—LYZL00000000) and the Czech Republic (11069/A, GAN—PRJNA728954) available in the GenBank database. The *aac(6′)-Ib-cr* gene confers resistance against both aminoglycosides and fluoroquinolones. More details about *A. baumannii* ST195^Ox^ isolates used in the comparative genomic analysis are listed in Appendix A.

**Table 1 life-11-01079-t001:** Phenotype and genotype and of the *Acinetobacter baumannii isolate 11069/A*.

Carbapenemases	ST^P/OX^	Additional *β*-Lactamases	Antibiotics (mg/L) ^a^							
			COT	MER	TIG	AMI	GEN	COL	CIP	TOB
OXA-23, OXA-58, OXA-66	2/195	ADC-73, TEM-1	>128	>16	0.25	>64	>32	2	>16	>32

Legend: MLST—multilocus sequence typing, MIC—minimum inhibitory concentration, COT—cotrimoxazole, MER—meropenem, TIG—tigecycline, AMI—amikacin, GEN—gentamicin, COL—colistin, CIP—ciprofloxacin, TOB—tobramycin. STP/Ox, sequence type according to Pasteur/Oxford multilocus sequence typing scheme. ^a^ Susceptible (S)/resistant (R) breakpoints (mg/L): COT, S ≤ 2, R > 4; MER, S ≤ 2, R > 8; TIG, insufficient evidence; AMI, S ≤ 8, R > 8; GEN, S ≤ 4, R > 4; COL, S ≤ 2, R > 2; CIP, S ≤ 0.001, R > 1; TOB, S ≤ 4, R > 4.

**Table 2 life-11-01079-t002:** Distribution of the resistance genes in *Acinetobacter baumannii* strain 11069/A.

Antibiotics	Resistance Gene	Identity (%)	Query/Template Length (bp)	Product	Accession Number
*β*-lactam	*bla* _ADC-73_	100	1152/1152	*β*-lactamase (AmpC)	CP050390
	*bla* _TEM-1_	100	861/861	*β*-lactamase (broad-spectrum)	CP050916
	*bla* _OXA-23_	100	822/822	*β*-lactamase (carbapenemase)	AY795964
	*bla* _OXA-58_	100	843/843	*β*-lactamase (carbapenemase)	AY665723
	*bla* _OXA-66_	100	825/825	*β*-lactamase (carbapenemase)	AY750909
Aminoglycoside	*aph(3′)-Ia*	100	816/816	Aminoglycoside phosphotransferase	X62115
	*aph(3″)-Ib*	100	804/804	Aminoglycoside phosphotransferase	MZ419555
	*aph(6)-Id*	100	837/837	Streptomycin phosphotransferase	MW690133
	*armA*	100	774/774	Aminoglycoside methyltransferase	AY220558
Macrolide	*msr*(E)	100	1476/1476	ABC-F type ribosomal protection protein	FR751518
	*mph*(E)	100	885/885	Macrolide 2′-phosphotransferase	CP077784
Sulphonamide	*sul2*	100	816/816	Dihydropteroate synthase	AY034138
Tetracycline	*tet*(B)	100	1218/1218	Tetracycline efflux pump	CP051474

## Data Availability

The genome sequence of the *A. baumannii* isolate sequenced as part of this study has been deposited in the GenBank under BioProject Number PRJNA728954.

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
