# Peer review of "Insights into the Resistome and Phylogenomics of a ST195 Multidrug-Resistant Acinetobacter baumannii Clinical Isolate from the Czech Republic"

_life, 2021, doi:10.3390/life11101079_

Round 1

Reviewer 1 Report

Dear Authors, 

the work is well written on a very hot topic, but probably you could change as case report, due to the fact that only one strain is effectively analyzed. 

But there are some revisions that should be done.

Some References are probably not up to date.

Figure 1 is really essential for the article? The in silico analysis of GNNF bacteria is the focus of the paper? You should work for better integration of this part if you confirm that is due to report.

Consider eliminating Fig. 1 and also could consider including some PCR figures related to the detection of drug-resistance genes.

Reviewer 2 Report

The current study describes the molecular resistance profile of a single clinical strain of A. baumannii. The isolate was found to be a resistant to a range of antibiotics as determined by MIC assay. These findings were corroborated using a combination of targeted PCR and whole genome sequencing to identify the resistance determinants likely to be responsible. 

The study is rather limited in scope, as only a single isolate is studied. The study is also not particularly novel (other than the strain under investigation). Other publications have characterised the resistome of a range of A. baumannii isolates, but that is not to say the publication is without merit.  All findings which help better characterise resistance in clinical isolates are useful. 

The study is relatively well written and appropriate methods have been applied.  My specific comments for the authors are below:

Materials and Methods, Bacterial strains – where were the reference strains obtained from? NCTC?

Table 2 – units are required for the MICs. It would also be useful to include the EUCAST breakpoint for each antibiotic tested.

Page 7: When describing the bla genes, the paper states “These genes were found to be co-expressed with other resistance genes” – How was this confirmed? Were transcriptional studies performed? Do the authors mean co-occurring?

Round 2

Reviewer 1 Report

Dear Authors, 

Thank you for your kindly reply. 

I think that this second revision is more suitable with case report.

I read your comments and your revisions and I think now the paper is suitable for publication.

Author Response

Details of the manuscript corrections are as follows –

Reviewer’s comments are in red colour.

Answers to the comments are in blue colour.

New things which are added, as per the reviewer’s suggestions, are highlighted in yellow colour in the manuscript.

  1. Title: to be improved ; may be it could read “Phylogenomics of a Czech ST195 intensive care unit multidrug-resistant Acinetobacter baumannii”  or something similar

Answer: Thanks to the reviewer for this proposal. The title has been modified.

  1. Section 2.2.: The amplified PCR products were sequenced by Sanger sequencing, and the sequences were blasted against GenBank database.

Comment: avoid the use of “blasted”; and give the version of blast which was used and the website.

Answer: Thanks to the reviewer for reporting this comment. The sentence has been rewritten.

  1. Section 2.2.

Comment: please re-write the following sentence: “Nine β-lactamase-producing reference strains strains….”

  • Answer: Thanks to the reviewer for reporting this comment. The sentence has been rewritten.

  1. Section 2.3: “The Institut Pasteur and Oxford MLST scheme (http://pubmlst.org/abaumannii/) was used to type the isolate of interest and was determined from the genome sequence data.”

Comment: please give the full reference, and not only the web sites.

  • Answer: Thanks to the reviewer for reporting this comment. The reference has been added.

  1. Section 2.3: BLDB database ; Comment: please add proper reference for this database.
  • Answer: Thanks to the reviewer for reporting this comment. The reference has been added.

  1. Section 2.3: “Plasmid replicon-associated genes were detected using Geneious against an internal database with plasmid replicons of A. baumannii.”
    Major comment: please add a reference for this database or add a suppl. file presenting its content in terms of plasmid-borne DNA targets.
  • Answer: Thanks to the reviewer for reporting this comment. The text has been modified.

  1. Section 2.3: please improve the following sentence (see suggestions) “(DNA) Repeat(s) regions, as well as insertion sequences (IS), were identified and analyzed using the Geneious Prime.”
    Comment: best website for IS analyses is : https://www-is.biotoul.fr/index.php; please indicate depth of the database used by Geneious Prime for perfoming IS searches.
  • Answer: Thanks to the reviewer for reporting this comment. Additional explanation was added.

  1. Table S2 – title should be improved e. g. may be it could read: “Table S2. Properties of Acinetobacter baumannii ST195 isolates used in the comparative genom(e) analyses”.
  • Answer: Thanks to the reviewer for reporting this comment. The title has been modified.

  1. Fig. S1: in the legend, when two primer pairs for a DNA target were defined, please indicate which one gave the PCR products.
  • Answer: Thanks to the reviewer for reporting this comment. Missing data have been added.

  1. Section 3.2 “BLAST analysis of the sequenced PCR amplicons confirmed that the isolate carried the resistance genes blaADC-like, blaTEM-like, blaOXA-23-like, blaOXA-51-like and blaOXA-58-like.”
    Comment: please indicate in a concise way the significance of these matches; length of the DNA sequence, identities, quality scores, etc
  • Answer: Thanks to the reviewer for reporting this comment. Sentences about Sanger sequencing were removed from the text. Match between PCR and WGS results is described in the discussion.

  1. Section 3.3: “BLASTn was used to analyze the close plasmid sequences.”
    Comment: to be re-worded e. g. BLASTn was used to detect related plasmid
    Sequences (in GenBank).
  • Answer: Thanks to the reviewer for reporting this comment. The sentence has been rewritten.

  1. Section 3.3 : « … of blaOXA-66, genes encoding N-acetyltransferase and helix-turn-helix transcriptional regulator were followed” / please change “followed “ by “found”

  • Answer: Thanks to the reviewer for reporting this comment. The sentence has been rewritten.

  1. Fig. 1; to increase the impact of this fig. ; please indicate on this figure if these ARG are likely to be functional and if so which resistance they would confer (by making a link with Table 1).
  • Answer: Thanks to the reviewer for reporting this comment. The Figure was modified.

  1. Section 3.3: “Unexpectedly, right end of the contig showed a high level of similarity (56.7% coverage, 100% identity) to a 20,139-bp armA/msr(E)/mph(E)-carrying plasmid (p2BJAB07104 [GenBank accession number, GAN - CP003907]) identified in A. baumannii.”
    Comment: such observation can increase the impact of the paper; please add a map to illustrate this point.
  • Answer: Thanks to the reviewer for reporting this comment. New figure has been added in the Supplementary Materials.

  1. Please improve legend of Fig. 1; e. g. Schematic map of the genomic regions of strain 11069/A encoding antibiotic resistance genes. … etc
  • Answer: Thanks to the reviewer for reporting this comment. Further data have been added.

  1. Please re-word the following sentences:

(a) “with no mutations observed throughout its and promoter sequences.” Suggestion : with no mutation  over its coding frame and promoter sequences.
(b) “was conducted in order to identify gene islands”; suggestion: “…
was conducted in order to identify genomic islands or regions of genomic plasticity (RGP).

  • Answer: Thank you for the comments. Sentences were modified.

  1. Please locate the ARG on the map of Fig. 2 (links with codes used in table 2 and Fig. 1 would be appreaciated but be careful with the nomenclature of the RGPs / GAPs and those of Fig. 1)
  • Answer: Thank you for this comment. Figure has been modified.

  1. Please add a table describing the main contents of the RGPs / Gaps that were observed (length, G+C content, IS ?, tRNA, integrase gene? ; anything new?? These observations can increase the impact of the paper.
  • Answer: Thank you for this comment. Additional data have been added.

  1. Fig. 3; add elements from Table S2 on this figure (country) + disease, antibiotic profile, etc
  • Answer: Thank you for this comment. Figure has been modified. Additional data have been added.

  1. Discussion: Synthax errors: isolate was “ST2/S195” should read ST195 /
  • Answer: Thank you this comment. The error has been repaired.

  1. Please, do not cite, in the discussion, Figures and tables shown in the results. However, adding a drawing (a figure) giving your vision of the genomic rearrangements or evolutive steps which led to strain 11069/A ARG profile would be very interesting, and would increase the impact of this short report. (this could be like a graphical summary of the main issues discussed in this section).
  • Answer: Thank you for these comments. Figure has been added.

  1. Conclusions; please correct the following syntax error: “In the present study, five different β-lactamase(s) and …”
  • Answer: Thank you for this comment. The sentence has been modified.